# Polyploids of Brassicaceae: Genomic Insights and Assembly Strategies

**DOI:** 10.3390/plants13152087

**Published:** 2024-07-27

**Authors:** Donghyun Jeon, Changsoo Kim

**Affiliations:** 1Department of Science in Smart Agriculture Systems, Chungnam National University, Daejeon 34134, Republic of Korea; jemdong@cnu.ac.kr; 2Department of Crop Science, Chungnam National University, Daejeon 34134, Republic of Korea

**Keywords:** whole-genome duplication, diversification, glucosinolate, sub-genome discrimination, polyploid

## Abstract

The Brassicaceae family is distinguished by its inclusion of high-value crops such as cabbage, broccoli, mustard, and wasabi, all noted for their glucosinolates. In this family, many polyploidy species are distributed and shaped by numerous whole-genome duplications, independent genome doublings, and hybridization events. The evolutionary trajectory of the family is marked by enhanced diversification and lineage splitting after paleo- and meso-polyploidization, with discernible remnants of whole-genome duplications within their genomes. The recent neopolyploidization events notably increased the proportion of polyploid species within the family. Although sequencing efforts for the Brassicaceae genome have been robust, accurately distinguishing sub-genomes remains a significant challenge, frequently complicating the assembly process. Assembly strategies include comparative analyses with ancestral species and examining k-mers, long terminal repeat retrotransposons, and pollen sequencing. This review comprehensively explores the unique genomic characteristics of the Brassicaceae family, with a particular emphasis on polyploidization events and the latest strategies for sequencing and assembly. This review will significantly improve our understanding of polyploidy in the Brassicaceae family and assist in future genome assembly methods.

## 1. Introduction

The Brassicaceae family, commonly known as Cruciferae, is a significant group within the angiosperms and belongs to the Brassicales order. This family includes 5 supertribes, 58 tribes, and 4140 species, covering a range of plants like annuals, biennials, and herbaceous perennials [1,2]. Moreover, the family Brassicaceae has a pivotal role in agriculture as a source of crucial crops such as cabbage, cauliflower, and mustard. Additionally, it is recognized as a genetic diversity hotspot, featuring unique secondary metabolites like the ‘mustard oil bomb’, which consists of glucosinolates (GSLs) and myrosinase enzymes. These metabolites are crucial for studying the interactions between biochemistry, natural selection, and gene evolution, especially in the context of herbivory defense mechanisms [3]. The abundant genetic resources of the family position it as an ideal model for research in flowering plant biology, facilitating investigations into plant development and the impact of ancient genome duplications.

Whole-genome duplications (WGDs) have significantly influenced the evolution of land plants, including angiosperms. These duplications not only foster the emergence of new traits and evolutionary innovations, but also contribute to increased trait variations within a clade [4]. Polyploidization, the process of acquiring multiple sets of chromosomes, has been a pivotal factor in the evolution of angiosperms. Each flowering plant experienced at least one, often multiple, ancient polyploidy events. About 15% of speciation events in angiosperms are associated with an increase in ploidy. Additionally, at some point during the diversification of the angiosperm crown group, 47% to 100% of flowering plant species can be linked to a polyploidy event [5]. The Brassicaceae family, diverging from other eudicots approximately 24.5~60 million years ago, is also no exception. The discovery of major WGDs, such as the At-α, At-β, and At-γ events in *Arabidopsis thaliana*, have had a pivotal role in identifying similar patterns across other angiosperm lineages [2,6,7,8]. This family underwent primary diversification during the Neogene period, from about 23 to 2.6 million years ago. It was also theorized that almost 50% of the taxa within the Brassicaceae family may have originated from recent polyploid events [9]. These species’ divergence and major polyploidy often align with periods of climate instability, suggesting that having multiple sets of chromosomes might offer an evolutionary advantage in rapidly changing environments. This is particularly evident in the evolutionary trajectory of the Brassicaceae family, underscoring the significant impact of polyploidization in fostering the diversity and resilience of this plant group.

The genomes of the Brassicaceae family have more polyploidy compared to other flowering plants. Distinguishing sub-genomes in the polyploidy of Brassicaceae has been challenging, mainly due to the complexities associated with polyploids. However, advancements in genomic technologies, such as long-read sequencing and chromosome capture methods, have substantially enhanced the analysis and identification of these sub-genomes. These technological improvements have led to more precise genomic assemblies at the pseudochromosome level, enriching our understanding of genomic sequencing and the evolutionary biology of plants [10,11,12]. However, distinguishing between subgenomes remains a challenge due to their similarities. Many reviews have been published on the importance of polyploidy in plant evolution and environmental adaptation [13,14,15]. However, despite the high rates of polyploidy within the Brassicaceae family, there have yet to be any reviews specifically focusing on this family. However, although there have been significant advancements in the latest assembly strategies for polyploid crop genomes since the publication of [16], there has yet to be any progress in documenting these developments. This review examines the specific characteristics and impacts of polyploidization within the Brassicaceae family, particularly emphasizing genomic assembly strategies for polyploid crops. It aims to provide a comprehensive framework that will guide future genomic assembly studies for polyploidy. By elucidating the complexities and innovative approaches involved in assembling polyploid genomes, this review offers essential insights that are expected to propel further advancements in the field of polyploidy in the Brassicaceae family.

## 2. Polyploidization of the Brassicaceae Family

The Brassicaceae family is renowned for its frequent hybridization events. However, this might be an underestimation, as it needs to fully consider older, clade-specific WGDs, which are often masked by subsequent diploidization processes (Figure 1). WGD events significantly characterize the evolutionary history of the Brassicaceae family and the broader Brassicales order [17]. The paleopolyploid genome of the Brassicaceae family is diversified into four or more major lineages, forming distinct monophyletic clades or tribes [18]. While some of these lineages and tribes retained their paleopolyploidization characteristics, some underwent younger WGDs, forming several new genera and tribes. Estimations indicate that approximately one-fourth of the tribes within the crucifer family experienced diversification after the post-At-α genome duplications [19]. Unlike the older paleopolyploid ones, these younger mesopolyploid genomes display different phylogenomic characteristics. Due to their more recent origin, mesopolyploid clades show a more significant karyological variability and have multiple base chromosome numbers, reflecting their independent diploidization events [20]. More recent mesopolyploidization events were identified in eleven of the fifty-one tribes within the Brassicaceae family. The family underwent eleven additional tribe-specific WGDs during their evolution in the Miocene and Pliocene epochs [21]. These additional WGD events potentially impacted around 130 genera, averaging 1.45 WGDs per genus in 35 million years. This rate is notably higher than the average for land plants [22,23]. The genomes that underwent mesopolyploidization exhibit extensive diploidization. Chromosomal rearrangements, reduction in genome size, and genome fractionation characterize this process. Despite these significant alterations, remnants of the duplicated genomic regions from these mesopolyploidization events are still detectable in the current genomic structures. Recently evolved polyploids, known as neopolyploids (such as *Brassica napus*), are distinguished by their larger genome size, elevated chromosome count, duplicated gene content, and existing diploid progenitors. Neopolyploidy is prevalent, evident in 43% of families. The proportion of neopolyploids in each tribe slightly correlates with the number of species [22,24,25]. This evolutionary complexity reflects the dynamic nature of the Brassicaceae genome over millions of years. This suggests that polyploidization is an ongoing evolutionary process within the Brassicaceae.

### 2.1. Diversification of Brassicaceae

The Brassicaceae family, known for its history of multiple instances of paleo- and mesopolyploidization in the past and the prevalence of neopolyploids, offers a potential model for studying how diversity evolves through polyploidization events. Studies have shown that WGDs, particularly older events like paleopolyploidization, may increase diversification rates, but this increase often occurs after a considerable delay. Diversification rates increased after nearly half of the nine ancient polyploidy events examined in flowering plants [26]. Understanding the genomic evolution in the Brassicaceae family involves recognizing the ‘Lag-Time Model’ that occurs between polyploidization events and the resulting diversification. This model is marked by a period of genomic adjustment, primarily through post-polyploid diploidization. During this phase, the genome underwent significant alterations after it had doubled. It is important to understand that diversification was not directly initiated by the genome doubling itself, but rather by the subsequent diploidization process. This delay suggests that WGDs are important contributors to diversification, but are not the sole driving factor [27]. Neopolyploidization is often considered a direct cause of speciation, acting as a crucial mechanism for generating new species within the Brassicaceae family. This process, involving the duplication of the entire genome, facilitates the creation of novel genomic arrangements that can lead to significant evolutionary innovations. Therefore, while both neopolyploidization and mesopolyploidization are essential for understanding the evolutionary dynamics within the Brassicaceae, their roles differ. In this context, genome doubling serves as an initial trigger or a stepping stone, setting the stage for further evolutionary processes that can lead to the diversification and emergence of new species within this rapidly evolving plant family [22,25,28].

The evolution of the Brassicaceae family is closely linked to significant geological and climatic changes during the Oligocene, Miocene, and Pliocene periods. The diversification of the Brassicaceae crown group began around 30 million years ago with the split of the Aethionemeae tribe from the rest of the family, coinciding with a global temperature decline and the formation of the Antarctic ice sheet in the Early Oligocene [29]. Most of the major branches within the family, except Aethionemeae, originated just before 20 million years ago, over a relatively brief period of about 0.69 million years. This period saw the emergence of the primary Lineages I, II, III, and IV. However, most tribal diversification happened after a significant drop in average temperatures that followed the Middle Miocene Climatic Optimum, which occurred between 18 and 14 million years ago [8]. These observations suggest that a global trend towards cooler and drier conditions may have influenced the diversification and expansion of the Brassicaceae.

### 2.2. Evolution of the Biosynthetic Pathways to Produce GSLs

GSLs are sulfur-rich, amino acid-derived metabolites predominantly found in the Brassicales order. These compounds have a basic structure comprising a sulfonated oxime or sulfated isothiocyanate group linked to thioglucose and a variable side chain [30]. GSLs, along with their activating enzymes, myrosinases, form a defense mechanism known as the mustard oil bomb. This mechanism is crucial for defense against pathogens and herbivores, acting through signaling, antimicrobial actions, and as deterrents [31]. The presence of specific proteins like epithiospecifier proteins and nitrile-specifier proteins significantly influences the transformation of GSLs into isothiocyanates or nitriles [32,33].

Before the At-β WGD event, GSLs were primarily synthesized from phenylalanine and branched-chain aliphatic amino acids through a core pathway regulated by an MYB–MYC complex. Post the At-β WGD, descendant plant families began synthesizing indole and methionine-derived GSLs. This development was accompanied by the retention of duplicate genes involved in regulation and early biosynthetic steps [34,35]. These duplicates underwent neofunctionalization, performing distinct roles and producing different substrates and products. For example, duplicated MYB genes specialized to function in individual GSL pathways, while MYC genes adapted to function in both pathways [36]. Following the At-α WGD, retained gene duplicates for enzymes like flavin monooxygenase glucosinolate S-oxygenase and indole GSL methyltransferase underwent tandem duplication and subfunctionalization. These processes enabled these enzymes to catalyze modifications of side chains in Met-derived GSLs and indole GSLs [37,38].

Regarding GSL transport within plant tissues, NRT1/PTR family proteins have a crucial role. The evolution of GSL transport likely predates the At-α WGD and initially involved NPF transporters with broad or nonspecific side chain affinity. Post-At-α WGD, these transporters evolved to transport specific GSL substrates, like indole GSLs, selectively [39]. Additionally, GSLs are structurally related to cyanogenic glucosides, widespread metabolites in plants and some arthropods. Phylogenetic analysis indicated that specific GSL transporters are orthologous to cyanogenic glucoside transporters found in cassava. This suggests that ancestral cyanogenic glucoside transporters evolved to develop a high affinity for cyanogenic glucosides and later for specific GSLs [40] through duplication and subfunctionalization. Additionally, genes associated with GSL biosynthesis tend to be preferentially retained or duplicated following polyploidization. These genes include ones encoding methylthioalkylmalate synthase enzymes, isopropylmalate isomerases, and branched-chain amino acid aminotransferases, which have been observed to either retain or duplicate in each subgenome post-allotetraploidization. Notably, the gene clusters responsible for encoding flavin-containing monooxygenases, crucial enzymes in sinigrin synthesis, exhibited a tendency for preferential retention following polyploidization. This cluster was predominantly expressed and underwent tandem duplication, contributing to a higher accumulation of sinigrin [41,42].

### 2.3. Polyploidy and Its Role in the Evolution and Domestication of Brassica Species for Agricultural Trait Development

Polyploidy, the condition of having more than two complete sets of chromosomes, is often linked to numerous desirable agricultural traits, such as larger seed size [43], increased stress tolerance [44], and enhanced disease resistance [45]. Recent phylogenetic analyses show that cultivated plants have experienced significantly more polyploidy events than their wild relatives. Often preceding domestication, lineages with recent WGDs are nearly twice as likely to be domesticated compared to those without WGDs [46].

In the genus Brassica, a clear example of the impact of polyploidy is evident in the classic ‘Triangle of U’, comprising six cultivated species derived from three diploid progenitors: *Brassica rapa*, *Brassica nigra*, and *Brrassica oleracea*. These diploids hybridized to form three distinct allopolyploid species—*Brassica carinata*, *Brassica juncea*, and *B. napus*—which then underwent biased gene loss and genome restructuring [47]. Artificial selection has repeatedly targeted various harvestable parts within these species, including underground storage organs (turnips), leafy vegetables (cabbages, pak choi), axillary buds (*Brussels sprouts*), floral parts (cauliflower, broccolini), and seeds (oilseeds) [48]. Studies on the Brassica Triangle reveal how polyploidy can disrupt highly conserved leaf structure-function relationships in seed plants, leading to novel trait correlations absent in parental species. These new genomic combinations and interactions break ancestral phenotypic trait correlations, creating opportunities for trait improvement. The genomic and chromosomal instability in early generations of allopolyploids enables the exploration of new trait spaces, potentially reaching higher adaptive peaks than their progenitors despite possible transient fitness costs.

A crucial part of understanding the domestication and adaptation processes in these allopolyploid Brassica species is recognizing the role of interploidy introgression, especially in *B. napus*. This study emphasizes how interploidy introgression was pivotal during the transition from winter to semi-winter rapeseed, aiding adaptation to the milder winter climates of Asia. Detailed genomic analyses identified introgression events from Asian-cultivated *B. rapa* morphotypes into semi-winter rapeseed, affecting between 2201 and 6950 genes involved in critical biological pathways such as developmental growth regulation, flower development, seed growth, lipid storage, glucosinolate catabolism, and gibberellin response. The flowering time-related genes were particularly significant, adapting to reduced vernalization requirements suitable for milder Asian winters. Key genes related to vernalization, circadian clocks, photoperiods, and autonomous pathways highlighted the vital role of interploidy introgression in adjusting flowering times to local conditions [49].

Furthermore, the creation of tetraploid radish plants using colchicine—validated by cytological observations and flow cytometry analysis of DNA content—demonstrates the ‘gigas’ effect, for which both vegetative and reproductive organs increase in size. These tetraploid radishes exhibited enhanced physiological traits, including increased antioxidant enzyme activity, soluble content, and overall quality. Notably, while some endogenous plant hormone levels exhibited negative trends, others, like abscisic acid and indoleacetic acid, positively correlated with delayed bolting and flowering, in contrast to diploid plants [50].

In addition, newly created artificial allotetraploid plants may provide a unique opportunity to create a new species without genetic modification technologies, although it is a very classical method. This suggests that polyploidy within the Brassicaceae not only fosters the emergence of new, agriculturally beneficial traits and offers an excellent target for artificial selection and crop improvement. The evolutionary instability of traits following hybridization makes them ideal for breeding programs to enhance crop resilience and productivity.

## 3. Challenges and Solutions in Sequencing and Assembling Polyploidy Plant Genomes

A significant number of polyploids have been identified as ancient or recent polyploids within the vast array of plant species. These plants possess multiple chromosome sets due to events like interspecific hybridization (allopolyploidy) or genome doubling (autopolyploidy) [51]. Particularly, the Brassicaceae family contains a large number of polyploid crops. The genome assembly of polyploids in the Brassicaceae family has seen limited progress, primarily focused on major crops (Table 1). Recently, a pan-genome platform for 86 accessions of *B. carinata* has been established, and an integrated platform combining the pan-genome and multi-omics data for *B. napus* has been developed [52,53]. Polyploidy introduces specific challenges at the pseudo-chromosome level or haplotype-resolved assembly. It involves complex tasks like assembling duplicated regions and distinguishing between inter-genomic and intra-genomic polymorphisms. Despite advances in sequencing technologies and computational algorithms over the past two decades, which have aided in analyzing genomes from nonvascular to flowering plants, assembling complex plant genomes remains daunting. Challenges such as high heterozygosity, repetitive sequences, and varied ploidy levels are difficult to address using traditional sequencing and assembly methods [54]. Therefore, to solve these problems, it is important to find an appropriate method for each genome characteristic.

### 3.1. Assessing Ploidy Levels in Plants

In genomic research focused on plants with varying ploidy levels, the crucial step is usually to ascertain the chromosome count or ploidy level most directly achieved by counting the somatic chromosomes in meristematic cells. Nevertheless, direct chromosome counting can be labor-intensive and time-consuming. It often necessitates the expertise of a researcher skilled in cytology, and in certain species, it is challenging due to factors like abundance, small size, or weak staining of the chromosomes. Additionally, understanding the genetic and morphological traits of polyploid plants is crucial. This understanding bridges the gap between sequencing challenges and practical applications, essential for accurate genome assembly and comprehending the evolutionary role of polyploidy [49].

Shifting into a polyploid state typically increases cell size, resulting in polyploids having larger cells than their diploid counterparts. This enlargement in cell size, particularly in the nucleus, can affect the overall structure and regulatory functions of the cell, reflecting the impact of ploidy changes on cellular architecture [60]. Consequently, these alterations might reduce metabolic and growth rates in polyploid organisms because of the longer replication time [61]. Efforts to assess ploidy led to the creation of various indirect methods for estimating ploidy. These methods utilize the relationship between genome copy number and the size or volume of cells or organs. Commonly used indicators of ploidy include measurements like stomatal guard cell length, stomatal frequency, chloroplast count in guard cells, or pollen diameter [62,63,64]. While these measures have proven effective as quick indicators of ploidy in many polyploid plant groups, they have limitations. These include sensitivity to environmental factors, the need for specific calibration, and overlapping measurement ranges across different ploidy levels. Such issues can lead to uncertainties and challenges when applying these methods to non-model plant species.

The ploidy level also can be determined using flow cytometry to estimate nuclear DNA content. Flow cytometry is an efficient, high-throughput technique that simultaneously measures multiple optical properties of individual particles, such as cells or nuclei, often tagged with fluorescent dyes. The data obtained are then utilized to deduce the physical or chemical properties of these particles, including genome size or ploidy. However, because flow cytometry assesses the total nuclear DNA content without considering chromosome count, determining the exact ploidy level necessitates calibration against a reference sample with a known chromosome number, typically established through conventional karyotyping. These morphological and genetic traits allow for the inference of plant polyploidy, further elucidated in genotypic analysis [65,66].

### 3.2. Genomic Characteristics of Polyploid Plants

The marker-based approach is crucial in revealing the complex genomic structures of polyploid plants. The first most apparent sign of polyploidy in marker genotyping is detecting multiple bands from putative single-locus markers. Instances such as observing more than two bands in single-locus assays like RFLPs or SSRs often indicate genome duplication in the target species [67,68]. For example, in *B. napus*, an allotetraploid species of recent origin, most SSR markers amplify both homologous loci, resulting in two to four bands per marker, varying with locus heterozygosity. Although whole genome duplications often result in multiple signals in genotyping assays, they are not the only cause. Other factors, like small-scale sequence duplications from non-homologous translocations, activation of retrotransposons, or unique aspects of primer sequences or enzyme restriction sites, can also lead to multiple responses. Therefore, observing several reactions in genotyping does not always indicate a whole genome duplication; thus, other potential causes should be considered [69]. However, suppose a large number of genotyping tests specific to certain genetic locations frequently detect multiple genome sequences or products. In that case, it strongly indicates the presence of polyploidy in the species being studied.

In genome profiling, k-mer frequencies in polyploid genomes have a crucial role. It enhances the modeling of polyploid genomes by taking the k-mer spectrum as input and using a mixture of negative binomial distributions to estimate genome size, repetitiveness, and heterozygosity rates. For instance, the diploid *A. thaliana* shows two significant peaks in its k-mer profile, whereas the triploid Meloidogyne enterology has three [70]. The heights of these peaks in the k-mer spectrum are proportional to the species’ heterozygosity. Additionally, the relative heights of the peaks within the k-mer spectrum indicate the level of heterozygosity present in a species. To illustrate, in the case of a diploid species, an increase in heterozygosity leads to a heightened first peak and a diminished second peak in the k-mer spectrum. This relationship is more intricate for polyploid species, but generally, heightened heterozygosity results in an elevated first peak and reduced subsequent peaks. Furthermore, the peaks with higher coverage in the k-mer spectrum signify the presence of increasingly duplicated repetitive sequences within the genomes. An important application of k-mer involves distinguishing between allotetraploid and autotetraploid species by analyzing distinct patterns of nucleotide heterozygosity rates. For instance, in wasabi, a well-documented phenomenon during meiosis is the formation of bivalents where homologous chromosomes from the same subgenome preferentially pair with each other (Figure 2). This phenomenon is also evident in various other allotetraploid species. Consequently, in allotetraploids, one would anticipate a higher proportion of “aabb” and a lower proportion of “aaab” because preferential pairing ensures the presence of two homologs from each of the first and second subgenomes after recombination. Conversely, in the case of potatoes, most cells exhibit quadrivalents during meiosis. In this scenario, an individual might possess 0, 1, 2, or 3 homologs from a given subgenome after recombination. Consequently, “aaab” is expected to be more prevalent than “aabb” because there are more likely to be either one or three copies of a subgenome rather than exactly two [70].

### 3.3. Sequencing Technology Advancements and Limitations in Polyploids

Reference genomes hold a pivotal role in the field of genomics and genetics. They serve as foundational blueprints for understanding the genetic makeup of organisms, both for individuals and entire species. However, in polyploid genomes, this becomes more complicated. The assembly of a polyploid genome resembles a compounded set of haplotype reconstruction problems, in which the computational complexity increases with higher ploidy [16]. The advancement of long-read sequencing technologies like PacBio and Oxford Nanopore Technologies has significantly eased the process of polyploidy sequencing compared to past methods. These technologies have revolutionized the ability to sequence complex genomes by providing longer reads, which offer a more comprehensive view of genomic structure and variation [71,72]. Additionally, the development of chromosome capture techniques such as Hi-C, Pore-C, and Omni-C has furthered our understanding of chromosomal organization and interaction within the nucleus [73,74]. Despite these advancements, distinguishing sub-genomes within polyploid species remains a challenging task. This complexity arises from the intricate nature of these genomes, where similar sequences across different sub-genomes make it challenging to assign specific sequences to their respective genomic origins accurately. The assembly of autopolyploid genomes is exceptionally challenging, as fragments of a sub-genome might be mistakenly assigned to the wrong sub-genome, resulting in incorrectly assembled genomes. Allopolyploids face similar challenges, but their greater genetic distance usually makes resolving their sub-genomes less complicated. This ongoing challenge in sub-genome discrimination highlights the need for more refined analytical tools and methods in genomic research [75].

### 3.4. Sub-Genome Discrimination in Allopolyploids

Historically, the identification and differentiation of subgenomes in various species relied on using general genomic features as markers. Characteristics like GC content, centromeric array sequences, microsatellites, and repetitive elements were used as ‘genomic signatures’ to categorize DNA sequences into subgenomes [76,77]. However, while these methods were effective in some instances, they also had notable limitations. This deficiency underscored the difficulties in subgenome phasing, especially within intricate genomic settings in which conventional genetic markers may not have provided the necessary precision for accurate distinction. Several alternative approaches have been proposed to address these challenges. Investigating species of similar lineages, particularly those in phylogenetically proximate areas, is crucial for detecting polyploidy and hybridization events. This approach is especially beneficial in species like Brassica, in which allopolyploid species such as *B. napus*, *B. juncea*, and *B. carinata* contain genomes similar to those of extant diploid species *B. rapa*, *B. oleracea*, and *Brassica nigra*. Including these diploid species in assays provides crucial insights into genome locations within the allopolyploids [21,78]. Based on the understanding that investigating closely related species, particularly those in the phylogenetically proximate area, is crucial for detecting polyploidy and hybridization events, a similar approach was applied to *Eutrema japonicum*. By comparing its genomic sequences with those of its diploid relatives, *Eutrema salsugineum* and *Eutrema yunnanense*, and previously sequenced *E. japonicum*, it was possible to distinguish the haplotypes of *E. japonicum* [58]. This approach, involving investigating closely related species, offers significant advantages in understanding polyploidy and hybridization (Figure 3). It enables a more precise identification of subgenomes, particularly in complex genetic landscapes like those of *Eutrema* spp. However, the method also has limitations, as it relies heavily on the availability and quality of genomic data from related species and may only sometimes provide comprehensive insight into more diverse or less studied genomes.

When genomic data from closely related species is unavailable, the k-mer and LTR-RT based method provides an effective alternative for distinguishing sub-genomes (Figure 4). K-mer phasing involves identifying specific repetitive DNA sequences to categorize subgenomes. This method uses scanning of k-mers and the k-means algorithm to cluster chromosomes into distinct subgenomes. Techniques like PCA and Student’s t-tests are employed to ensure the accuracy of this phasing. Additionally, this approach includes mapping specific k-mers to the genome and applying Fisher’s exact test to confirm their significance. A key focus is on LTR-RTs, which are prevalent in plant genomes and act as markers to estimate the timeframe from subgenome differentiation to allohybridization [79,80]. This comprehensive approach, which integrates multiple bioinformatics techniques, is crucial for accurately distinguishing and understanding subgenomes, especially in cases in which closely related genomic data are not readily available [80]. However, it is important to note that these methods are primarily effective for allotetraploids, and may not be as helpful in distinguishing autotetraploids.

### 3.5. Genome Assembly Strategies of Autopolyploids in the Brassicaceae

In the Brassicaceae family, autopoliploidy was traditionally thought to be uncommon. However, recent reports identified plants like *Capsella bursa-pastoris* as naturally occurring autotetraploids. While not yet widely reported, it is suspected that there are several other autopolyploids within this family [81]. Autopolyploids present unique challenges in genome assembly due to the presence of four similar haplotypes with a significant proportion of nearly identical sequences. One major difficulty in assembling autoployploid genomes is the tendency to link these identical sequences, often leading to chimeric contigs with switch errors or false duplications. These chimeric contigs can interfere with Hi-C signals, causing erroneous scaffolds that mix sequences from different haplotypes. Additionally, differentiating nearly identical homologous sequences across haplotypes is challenging, often resulting in many collapsed contigs. These collapsed contigs can create misleading Hi-C links between the phased contigs of different haplotypes, leading to incorrect and overly long scaffolds. Despite these challenges, recent advancements in autotetraploid genome assembly were made using innovative techniques like single-cell sequencing of pollen genomes combined with Hi-C data. This approach, known as ‘gamete-binning’, begins with sequencing long reads from somatic DNA to create an initial contig-level assembly. Simultaneously, gamete genome sequencing data were collected, grouping contigs into clusters representing individual haplotypes based on genetic linkage. Long reads were then assigned to haplotypes based on their similarity to these contigs. This method enables a separate assembly of each haplotype, and facilitates chromosome-scale scaffolding using Hi-C data [82,83]. While not yet applied to the Brassicaceae family, this technique shows great potential for advancing genome assembly in autopolyploid crops in the Brassicaceae family.

## 4. Conclusions

The Brassicaceae family, characterized by extensive polyploidization, showcases evolutionary complexity through its hybridization events and WGDs. These events led to a significant diversification within the family, creating a range of polyploid species with varying genomic architectures. Despite the challenges in genome assembly and sub-genome discrimination due to polyploid complexity, advancements in sequencing technologies and methodologies, including ‘gamete-binning’ and k-mer analysis, have improved our understanding of polyploid genomes. These tools offer new insights into the genetic diversity and evolutionary history of the Brassicaceae family, highlighting the ongoing need for innovative research strategies to tackle the complexities of polyploid genomes. This review offers valuable insights into the complexities of the Brassicaceae genome, highlighting the evolutionary significance of polyploidization. By underscoring the potential for genetic advancements in crop improvement and emphasizing the need for ongoing plant genomics research, this work lays the groundwork for future agricultural innovations within the Brassicaceae family through its comprehensive exploration of genomic diversity.

## Figures and Tables

**Figure 1 plants-13-02087-f001:**
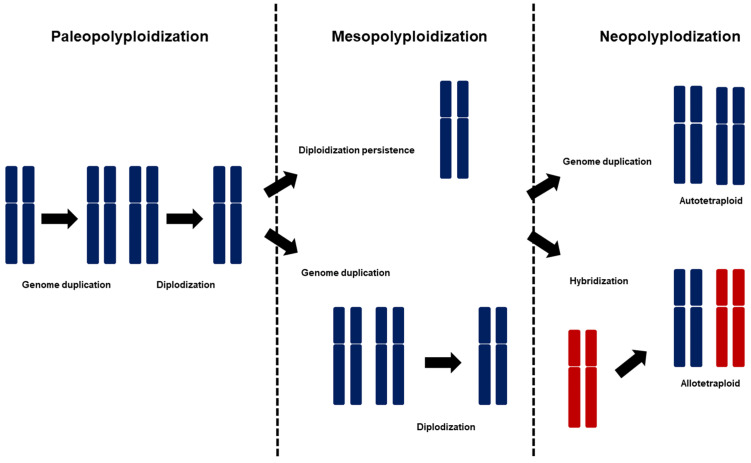
The evolutionary procedure of polyploidization in the Brassicaceae family. The blue symbols represent diploids that have undergone the conventional evolutionary process, while the red symbols denote closely related diploids. The three phases of polyploidization: paleopolyploidization, mesopolyploidization, and neopolyploidization. In paleopolyploidization, a genome duplication event is followed by diploidization, where the duplicated genome returns to a diploid-like state over evolutionary time. In mesopolyploidization, a genome that was diploidized during paleopolyploidization undergoes another genome duplication event and is diploidized once again. Alternatively, the diploid genome may remain unchanged without undergoing further duplication. In neopolyploidization, polyploidy is maintained after experiencing genome duplication. The genome can be fully replicated to become an autotetraploid or hybridize with a closely related species to form an allotetraploid.

**Figure 2 plants-13-02087-f002:**
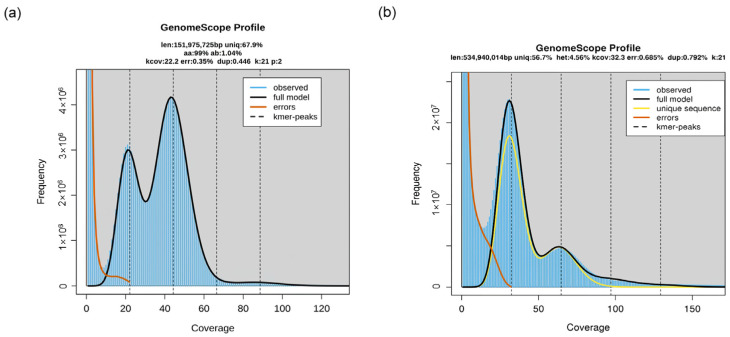
The GenomeScope plots for the (**a**) diploid *Arabidopsis thaliana* [70] and (**b**) allopolyploid *Eutrema japonicum* within the Brassicaceae family.

**Figure 3 plants-13-02087-f003:**
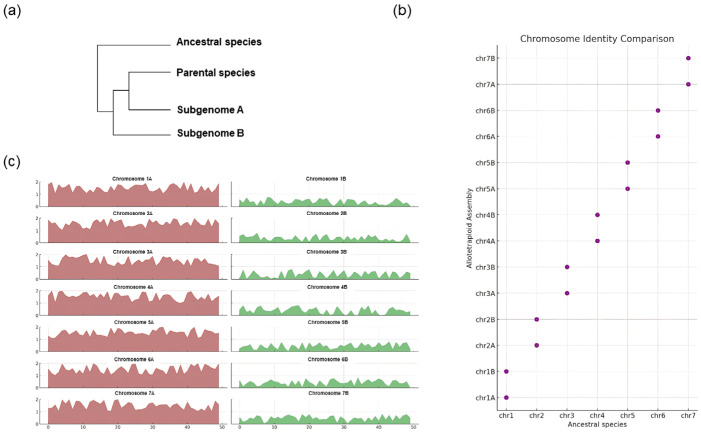
Subgenome discrimination using genomic information from closely related species. (**a**) The phylogenetic tree of the relationship between subgenome A, B, and its closely related species. (**b**) Distinguishing subgenomes through a comparison of sequence similarity with ancestral species. (**c**) Mapping analysis against parental species, with colored regions indicating mapping coverage. This Figure was reproduced based on the data from Tanaka et al. [58].

**Figure 4 plants-13-02087-f004:**
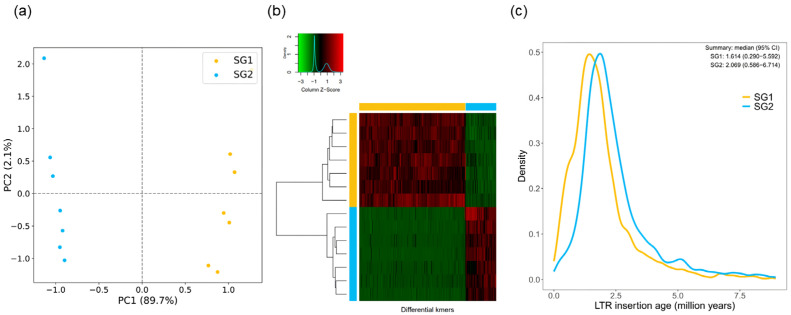
Subgenome phasing of allotetraploid *Eutrema japonicum*. (These Figures were from unpublished data). (**a**) A Principal Component Analysis (PCA) confirming effective phasing into three distinct subgenomes, evident by the unique patterns of differential 15-mers and homoeologous chromosomes. (**b**) A hierarchical clustering approach, where the top axis color bar represents k-mer specificity to each subgenome and the side axis color bar denotes chromosome assignment to subgenomes, with the heatmap reflecting Z-scaled k-mer abundance. (**c**) The timeline of LTR-RT insertions specific to each subgenome.

**Table 1 plants-13-02087-t001:** The current status of polyploid genome assembly within the Brassicaceae family.

Species	Estimated Genome Size	Ploidy Level	Long-Read Sequencing Platform	Chromosome Capture Sequencing Platform	Ref.
*Arabidopsis suecica*	~272 Mb	Allotetraploid	PacBio Sequel	Hi-C	[55]
*Armoracia rusticana*	~636 Mbp	Allotetraploid	ONT, PacBio HiFi	Hi-C	[12]
*Brassica carinata*	~1.31 Gbp	Allotetraploid	PacBio	Hi-C	[47]
*Brassica juncea*	~922 Mb	Allotetraploid	PacBio RSII	Hi-C	[10]
*Brassica napus*	~1132 Mb	Allotetraploid	PacBio SMRT	Hi-C	[56]
*Cardamine enshiensis*	~443 Mb	Allotetraploid	PacBio Sequel	Hi-C	[57]
*Eutrema japonicum*	~1512.1 Mb	Allotetraploid	PacBio CLR	Hi-C	[58]
*Pugionium cornutum*	~570 Mb	Allotetraploid	GridION, PacBio RS II		
*Pugionium dolabratum*	~606 Mb	Allotetraploid	GridION, PacBio RS II		[59]

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
