# Peer review of "Polyploids of Brassicaceae: Genomic Insights and Assembly Strategies"

_plants, 2024, doi:10.3390/plants13152087_

Round 1

Reviewer 1 Report

Comments and Suggestions for Authors

When I read the title of the review I thought that I was going to read a bioinformatics review on assembly strategies. This was certainly not the case. There is no bioinformatics content whatsoever, but only some low depth description on techniques used for assembling genomes, like e.g. the availability of PacBio or OxfordNanopore technology. So either the title is adjusted in a way that it fits with the content of the review or the content of the review is adjusted which is considerably more work.

In general the review does not contain a large amount of in depth information but is a rather shallow description of aspects linked to polyploid genomes. It is not a bad review, it is just not very broad.

More specific comments:

In their general introduction the authors mention the number of tribes etc. This is an outdated number. The authors should cite

German DA, Hendriks KP, Koch MA, Lens F, Lysak MA, Bailey CD, Mummenhoff K, Al-Shehbaz IA (2023) An updated classification of the Brassicaceae (Cruciferae). PhytoKeys 220: 127-144. https://doi.org/10.3897/phytokeys.220.97724

as well as

Hendriks KP, Kiefer C, Al-Shehbaz IA, Bailey CD, Hooft van Huysduynen A, Nikolov LA, Nauheimer L, Zuntini AR, German DA, Franzke A, Koch MA, Lysak MA, Toro-Núñez Ó, ÖzüdoÄŸru B, Invernon VR, Walden N, Maurin O, Hay NM, Shushkov P, Mandáková T, Thulin M, Windham MD, Rešetnik I, Španiel S, Ly E, Pires JC, Harkess A, Neuffer B, Vogt R, Bräuchler C, Rainer H, Janssens SB, Schmull M, Forrest A, Guggisberg A, Zmarzty S, Lepschi BJ, Scarlett N, Stauffer FW, Schönberger I, Heenan P, Baker WJ, Forest F, Mummenhoff K, Lens F (2022) Less is more: global Brassicaceae phylogeny based on filtering of 1,000 gene dataset. BioRxiv, 1–44. https://doi.org/10.1101/2022.09.01.506188 [preprint, posted September 07, 2022]

in that paragraph.

Further, the authors write that the estimated age of the Brassicaceae is 60 my. The authors should check more sources because divergence time estimates are not trivial and there are quite a number of different estimates around. These should be mentioned, particularly as the authors try to put these dates into relation to climatic events.

I am missing a citation for the statement from line 98 to 99 on the higher rate than the average rate for land plants.

Genus and taxon names should always be in italics.

Comments on the Quality of English Language

The manuscript reads well and there are no obvious errors.

Author Response

Comments and Suggestions for Authors

When I read the title of the review I thought that I was going to read a bioinformatics review on assembly strategies. This was certainly not the case. There is no bioinformatics content whatsoever, but only some low depth description on techniques used for assembling genomes, like e.g. the availability of PacBio or OxfordNanopore technology. So either the title is adjusted in a way that it fits with the content of the review or the content of the review is adjusted which is considerably more work.

In general the review does not contain a large amount of in depth information but is a rather shallow description of aspects linked to polyploid genomes. It is not a bad review, it is just not very broad.

-

Response to Reviewer

Thank you for your insightful comments. We appreciate your observations and agree with your points. As researchers deeply involved in the practical process of genome assembly and studying genome evolution, particularly focusing on the Brassicaceae family and polyploid crops, our review reflects the challenges and inspirations we encountered in our work.

We understand that there are numerous reviews covering sequencing technologies, but we noticed a significant gap in reviews focusing specifically on assembly strategies, particularly updated information on recent technologies. Hence, we chose to concentrate on subgenome assignment and assembly strategies, emphasizing polyploidy. This focus led us to create a specialized review, targeting the unique challenges and updates related to polyploidy in the Brassicaceae family.

While our review may not provide in-depth information on bioinformatics per se, we aimed to differentiate it by addressing specific aspects of polyploid genome assembly that are less commonly covered. We appreciate your constructive feedback and will consider your suggestions to either adjust the title or expand the content to better meet the expectations for a comprehensive review on this topic.

Comments and Suggestions for Authors

More specific comments:

In their general introduction the authors mention the number of tribes etc. This is an outdated number. The authors should cite

German DA, Hendriks KP, Koch MA, Lens F, Lysak MA, Bailey CD, Mummenhoff K, Al-Shehbaz IA (2023) An updated classification of the Brassicaceae (Cruciferae). PhytoKeys 220: 127-144. https://doi.org/10.3897/phytokeys.220.97724

as well as

Hendriks KP, Kiefer C, Al-Shehbaz IA, Bailey CD, Hooft van Huysduynen A, Nikolov LA, Nauheimer L, Zuntini AR, German DA, Franzke A, Koch MA, Lysak MA, Toro-Núñez Ó, ÖzüdoÄŸru B, Invernon VR, Walden N, Maurin O, Hay NM, Shushkov P, Mandáková T, Thulin M, Windham MD, Rešetnik I, Španiel S, Ly E, Pires JC, Harkess A, Neuffer B, Vogt R, Bräuchler C, Rainer H, Janssens SB, Schmull M, Forrest A, Guggisberg A, Zmarzty S, Lepschi BJ, Scarlett N, Stauffer FW, Schönberger I, Heenan P, Baker WJ, Forest F, Mummenhoff K, Lens F (2022) Less is more: global Brassicaceae phylogeny based on filtering of 1,000 gene dataset. BioRxiv, 1–44. https://doi.org/10.1101/2022.09.01.506188 [preprint, posted September 07, 2022]

in that paragraph.

-

Response to Reviewer

Thank you for providing the latest research results. This information greatly enhances the credibility of our review. We genuinely appreciate it. We have added and replaced the references you suggested, and updated the introduction in the main text to: "This family includes five supertribes, 58 tribes, and 4,140 species, covering a range of plants like annuals, biennials, and herbaceous perennials [1,2]."

German, D.A.; Hendriks, K.P.; Koch, M.A.; Lens, F.; Lysak, M.A.; Bailey, C.D.; Mummenhoff, K.; Al-Shehbaz, I.A. An updated classification of the Brassicaceae (Cruciferae). Phytokeys 2023, 127-144, doi:10.3897/phytokeys.220.97724. 2.             

Hendriks, K.P.; Kiefer, C.; Al-Shehbaz, I.A.; Bailey, C.D.; van Huysduynen, A.H.; Nikolov, L.A.; Nauheimer, L.; Zuntini, A.R.; German, D.A.; Franzke, A.; et al. Global Brassicaceae phylogeny based on filtering of 1,000-gene dataset. Curr Biol 2023, 33, 4052-+, doi:10.1016/j.cub.2023.08.026.

Comments and Suggestions for Authors

Further, the authors write that the estimated age of the Brassicaceae is 60 my. The authors should check more sources because divergence time estimates are not trivial and there are quite a number of different estimates around. These should be mentioned, particularly as the authors try to put these dates into relation to climatic events.

-

Response to Reviewer

In response to the reviewer's suggestion, we have reviewed various divergence time estimates for Brassicaceae and found reliable references indicating a range of 24.5 to 60 million years ago. We have updated the manuscript accordingly. The revised references are as follows:

  • Hendriks, K.P.; Kiefer, C.; Al-Shehbaz, I.A.; Bailey, C.D.; van Huysduynen, A.H.; Nikolov, L.A.; Nauheimer, L.; Zuntini, A.R.; German, D.A.; Franzke, A.; et al. Global Brassicaceae phylogeny based on filtering of 1,000-gene dataset. Curr Biol 2023, 33, 4052-+, doi:10.1016/j.cub.2023.08.026.
  • Kagale, S.; Robinson, S.J.; Nixon, J.; Xiao, R.; Huebert, T.; Condie, J.; Kessler, D.; Clarke, W.E.; Edger, P.P.; Links, M.G.; et al. Polyploid Evolution of the Brassicaceae during the Cenozoic Era. Plant Cell 2014, 26, 2777-2791, doi:10.1105/tpc.114.126391.
  • Huang, X.C.; German, D.A.; Koch, M.A. Temporal patterns of diversification in Brassicaceae demonstrate decoupling of rate shifts and mesopolyploidization events. Ann Bot-London 2020, 125, 29-47, doi:10.1093/aob/mcz123.

Thank you for bringing this to our attention.

Comments and Suggestions for Authors

I am missing a citation for the statement from line 98 to 99 on the higher rate than the average rate for land plants.

-

Response to Reviewer

We have respectfully addressed the reviewer's concern by adding two references to support the statement from lines 98 to 99 regarding the higher rate than the average rate for land plants. The added references are

Hohmann, N.; Wolf, E.M.; Lysak, M.A.; Koch, M.A. A Time-Calibrated Road Map of Brassicaceae Species Radiation and Evolutionary History. Plant Cell 2015, 27, 2770-2784, doi:10.1105/tpc.15.00482. Walden, N.; German, D.A.; Wolf, E.M.; Kiefer, M.; Rigault, P.; Huang, X.C.; Kiefer, C.; Schmickl, R.; Franzke, A.;

Neuffer, B.; et al. Nested whole-genome duplications coincide with diversification and high morphological disparity in Brassicaceae. Nat Commun 2020, 11, doi:10.1038/s41467-020-17541-6. Thank you for pointing out this oversight.

Comments and Suggestions for Authors

Genus and taxon names should always be in italics.

-

Response to Reviewer

During the process of transferring the manuscript to the MDPI format, the italicization of genus and taxon names was inadvertently removed. This formatting issue escaped our initial notice, and we apologize for any confusion this may have caused. We have thoroughly reviewed the manuscript and corrected all instances where genus and taxon names were not properly italicized. We have ensured that all such names are now consistently and correctly formatted according to the standard conventions. Thank you for bringing this to our attention, and we appreciate your understanding.

Thank you for your thorough and precise feedback. Your insights have been invaluable in enhancing the quality and accuracy of our review. We sincerely appreciate your constructive comments and suggestions.

Reviewer 2 Report

Comments and Suggestions for Authors

The review by Jeon and Kim aims to examine the specific characteristics and impacts of polyploidization within the Brassicaceae family, particularly emphasizing genomic assembly strategies for polyploid crops, and providing a comprehensive framework that will guide future genomic assembly studies for plant polyploidy. Generally, I think this review offers essential insights that are expected to propel further advancements in the field of poly ploidy in the Brassicaceae family. Totally, I agree with this publication with absolute necessity in Plants. Please carefully check the word spellings and grammars to make sure them in the right forms before the acceptance.

Author Response

Comments and Suggestions for Authors

The review by Jeon and Kim aims to examine the specific characteristics and impacts of polyploidization within the Brassicaceae family, particularly emphasizing genomic assembly strategies for polyploid crops, and providing a comprehensive framework that will guide future genomic assembly studies for plant polyploidy. Generally, I think this review offers essential insights that are expected to propel further advancements in the field of poly ploidy in the Brassicaceae family. Totally, I agree with this publication with absolute necessity in Plants. Please carefully check the word spellings and grammars to make sure them in the right forms before the acceptance.

Response to Reviewer

Thank you very much for your thoughtful comments and suggestions on our manuscript. We are pleased to hear that you found our review valuable and believe it will contribute significantly to the field of polyploidy in the Brassicaceae family.

We appreciate your recommendation to carefully check the word spellings and grammar. We will thoroughly proofread the manuscript to ensure that all spellings and grammatical structures are correct before the final acceptance.

Thank you once again for your positive feedback and constructive suggestions. We look forward to the opportunity to contribute to the advancements in plant polyploidy studies.

Reviewer 3 Report

Comments and Suggestions for Authors

This review introduced the polyploidy characteristics of Brassicaceae family and the assembly strategies for polyploid genomes. The followings were suggested for authors in revising the manuscript.

1. Lines 45-46 showed ‘Approximately 15% of speciation events in flowering plants are due to polyploidization [4]’. Actually, reference [4] emphasize ‘15% of angiosperm speciation events are accompanied by ploidy increase’, and the authors then cited a paper which indicated ‘47% to 100% of flowering plant species can be traced to a polyploid event at some point within the diversification of the angiosperm crown group’. Please check the percentage of polyploids in flowering plants.

2. It is suggested to put ‘3.6. Polyploidy and Its Role in the Evolution and Domestication of Brassica Species for Agricultural Trait Development’ into ‘2. Polyploidization of the Brassicaceae Family’, Thus, the information about Brassicaceae family will be the focus of the first half part of this manuscript.

3. Some pan-genome studies on allopolyploid species in Brsssica species should be reviewed in this manuscript.

4. The authors should pay more attentions to the writing of plant scientific names, such as line 49 ‘Arabidopsis thaliana’, line 104 ‘Brassica napus’, line 251 ‘B. napus’, line 266 ‘A. thaliana’, line 290 ‘Arabidopsis thaliana’, line 291 ‘Eutrema japonicum’, line 327 ‘B. napus, Brassica juncea, and Brassica carinata’, line 328 ‘Brassica rapa, Brassica oleracea, and Brassica nigra’, line 334 ‘Eutrema salsugineum, Eutrema yunnanense, E. japonicum’, line 372 ‘Capsella bursa-pastoris’.

Author Response

Comments and Suggestions for Authors

This review introduced the polyploidy characteristics of Brassicaceae family and the assembly strategies for polyploid genomes. The followings were suggested for authors in revising the manuscript.

  1. Lines 45-46 showed ‘Approximately 15% of speciation events in flowering plants are due to polyploidization [4]’. Actually, reference [4] emphasize ‘15% of angiosperm speciation events are accompanied by ploidy increase’, and the authors then cited a paper which indicated ‘47% to 100% of flowering plant species can be traced to a polyploid event at some point within the diversification of the angiosperm crown group’. Please check the percentage of polyploids in flowering plants.

-

Response to Reviewer

Thank you for your valuable feedback and for pointing out the need for clarification regarding the percentages related to polyploidy in flowering plants. After reviewing the cited references, we have made the necessary corrections to ensure accuracy. The statement has been revised to: "About 15% of speciation events in angiosperms are associated with an increase in ploidy. Additionally, at some point during the diversification of the angiosperm crown group, 47% to 100% of flowering plant species can be linked to a polyploidy event." We believe this revision more accurately reflects the findings of the referenced studies and provides a clearer understanding of the role of polyploidy in the evolution of angiosperms. We appreciate your attention to detail and your constructive comments, which have helped improve the quality and accuracy of our manuscript. Thank you once again for your insightful feedback.

Comments and Suggestions for Authors

  1. It is suggested to put ‘3.6. Polyploidy and Its Role in the Evolution and Domestication of Brassica Species for Agricultural Trait Development’ into ‘2. Polyploidization of the Brassicaceae Family’, Thus, the information about Brassicaceae family will be the focus of the first half part of this manuscript.

-

Response to Reviewer

We appreciate the reviewer’s insightful suggestion. After careful consideration, we agree that moving the section ‘3.6. Polyploidy and Its Role in the Evolution and Domestication of Brassica Species for Agricultural Trait Development’ to the earlier section ‘2. Polyploidization of the Brassicaceae Family’ will create a more cohesive and logically structured manuscript. This change places the information about the Brassicaceae family at the forefront, ensuring that it is the primary focus of the first half of the manuscript. Accordingly, we have relocated this content to section 2.3. We believe this adjustment enhances the overall clarity and flow of our review. Thank you for your valuable input, which has significantly contributed to improving our manuscript.

Comments and Suggestions for Authors

  1. Some pan-genome studies on allopolyploid species in Brsssica species should be reviewed in this manuscript.

-

Response to Reviewer

We acknowledge the reviewer's suggestion to include pan-genome studies on allopolyploid species in Brassica. In the revised manuscript, we have added the following information under section 3. Challenges and Solutions in Sequencing and Assembling Polyploid Plant Genomes:

"The genome assembly of polyploids in the Brassicaceae family has seen limited progress, primarily focused on major crops (Table 1). Recently, a pan-genome platform for 86 accessions of B. carinata has been established, and an integrated platform combining the pan-genome and multi-omics data for B. napus has been developed [53,54]."

These additions can be verified through the track changes feature in the Word document.

Cui, X.B.; Hu, M.; Yao, S.L.; Zhang, Y.Y.; Tang, M.Q.; Liu, L.J.; Cheng, X.H.; Tong, C.B.; Liu, S.Y. BnaOmics: A comprehensive platform combining pan-genome and multi-omics data from B. napus. Plant Commun 2023, 4, doi100609 10.1016/j.xplc.2023.100609.

Niu, Y.; Liu, Q.Q.; He, Z.S.; Raman, R.; Wang, H.; Long, X.X.; Qin, H.; Raman, H.; Parkin, I.A.P.; Bancroft, I.; et al. A Brassica carinata pan-genome platform for Brassica crop improvement. Plant Commun 2024, 5, doi100725 10.1016/j.xplc.2023.100725.

These additions provide a comprehensive overview of the latest advancements in pan-genome studies within the Brassicaceae family, highlighting the significant progress made in understanding and utilizing the genetic diversity of allopolyploid species.

Comments and Suggestions for Authors

  1. The authors should pay more attentions to the writing of plant scientific names, such as line 49 ‘Arabidopsis thaliana’, line 104 ‘Brassica napus’, line 251 ‘B. napus’, line 266 ‘A. thaliana’, line 290 ‘Arabidopsis thaliana’, line 291 ‘Eutrema japonicum’, line 327 ‘B. napus, Brassica juncea, and Brassica carinata’, line 328 ‘Brassica rapa, Brassica oleracea, and Brassica nigra’, line 334 ‘Eutrema salsugineum, Eutrema yunnanense, E. japonicum’, line 372 ‘Capsella bursa-pastoris’.

-

Response to Reviewer

During the process of transferring the manuscript to the MDPI format, the italicization of genus and taxon names was inadvertently removed. This formatting issue escaped our initial notice, and we apologize for any confusion this may have caused. We have thoroughly reviewed the manuscript and corrected all instances where genus and taxon names were not properly italicized. We have ensured that all such names are now consistently and correctly formatted according to the standard conventions. Thank you for bringing this to our attention, and we appreciate your understanding.
